# An Introduction to Spiral Steroids

**DOI:** 10.3390/ijms23179523

**Published:** 2022-08-23

**Authors:** Fred Chasalow

**Affiliations:** 1IOMA LLC, Belmont, CA 94002-3321, USA; fchasalow@gmail.com; Tel.: +1-650-576-1800; 2Department of Laboratory Medicine, VAMC, San Francisco, CA 94121-1545, USA

**Keywords:** spiral steroids, ionotropin, digoxin-like materials, NaK-ATPase, potassium sparing diuretics

## Abstract

In addition to classical steroids, which have cholesterol as a precursor, there are steroids with 7-dehydrocholesterol as a precursor. This review describes the identification of these steroids, their biosynthesis, and some aspects of their function. There are three classes of these compounds, distinguished by the number of their carbon atoms, 23, 24, and 25. Each class has a spiral steroid and is a phosphodiester. Up until these investigations, no spiral steroids or steroid phosphodiesters were known. There are at least 13 compounds, of which six have been purified to near homogeneity; each one has been characterized by its mass and proposed composition, and they function by regulating the NaK-ATPase. Based on the tissues in which they have been detected, each class of compound seems to regulate a different isoform of the NaK-ATPase. This is an important site of endocrine regulation.

## 1. Introduction

Intracellular fluids have K+ levels of 100 mM. Extracellular fluids, including plasma, have K+ levels of 4–6 mM. Thus, there must be a mechanism to ‘pump’ K+ into cells against that gradient. In the 1950’s, Szent-Gyorgyi proposed that there was an endogenous equivalent to digoxin [1]. His speculation unleashed a 60-year search for the material, which he proposed might function similarly to digoxin. There have been several false candidates, primarily digoxin itself, ouabain, and more recently, marinobufagenin [2,3,4]. However, in vertebrates, no precursors or metabolites were identified for any of the candidates. Within recent years, we have identified 13 steroid phosphocholine diesters. These can be divided into three classes, based on the number of carbon atoms in the steroid fragment, and all can be derived from a common precursor. Some, but not all, bind to digoxin specific antibodies. In summary, the discovery points to a new path of investigation to understand K+ regulation. This has consequences for many diseases, especially diseases of pregnancy.

### 1.1. What Is a Spiral Steroid?

A carbon atom can have four substituents. If each pair is part of a ring, it would be a spiral steroid. The spiral carbon atom would be a bridge between the rings, but no bond would be part of both rings. What distinguishes a spiral steroid from other bridge carbons is that the planes formed by the two rings are perpendicular. All four of the substituents need not be carbon atoms. To date, carbon atom 17 is the only natural, spiral carbon atom known to be present in steroid molecules. Some synthetic potassium sparing diuretics, including spironolactone and eplerenone, are also spiral steroids. Figure 1 illustrates these features.

### 1.2. Naming, Symbols and Abbreviations

Other than the apparent precursor, 17α-hydroxy-pregna-5,7-dienolone, none of these compounds had been described [5,6]; thus, there are no widely recognized, common names. Chasalow selected *IONOTROPIN* as the name of the first spiral steroid isolated, as it seemed to be needed for regulation of ion transport [7]. In retrospect, kaleotropin might have been a better selection. All of the papers published from this laboratory have included Ionotropin as a keyword. This practice has made it possible for investigators to find all of the original papers.

There are no common names for steroids with 23, 24, or 25, carbon atoms. Rather than using cross reaction with an antibody specific to a particular cardiotonic glycoside as the basis for the symbol, we established the practice of using the molecular mass of the steroid fragment as observed in the positive ion mass spectrum, [8]. Note that the apparent mass of a fragment might not be the same as the true formula mass.

Finally, all of the compounds are phosphodiesters. For phosphocholine compounds we preface the observed mass with C; for phosphoethanolamine compounds, the preface is E; if the phosphodiester was not identified, the letter is X [9].

### 1.3. Mass Spectra of Steroid Phosphocholine Diesters

There must be two breaks in the gonane portion of a steroid in order to generate fragments. As a consequence, steroids usually generate simple mass spectra. In contrast to classical steroid hormones, phosphodiesters usually have more than one molecular ion present, depending on the counter cation associated with the phosphate [7]. For the phosphocholine diester, the biggest ion peak is the fragment derived from phosphocholine at *m*/*z* = 184 Da.

Each phosphocholine steroid diester can generate five different *m*/*z* cations, which were designated S, A, B, C, and D [10]. Figure 2 gives an example and Table 1 shows the positive ions that derive from each of the phosphocholine diesters. The relative amounts of S, A, B, C, and D, are dependent on [i] the source of the sample, [ii] the electrolytes present, and [iii] how the sample has been extracted or purified. This pattern has also been useful in identifying steroid phosphodiesters in crude extracts [10]. A further complication occurs as the phosphoethanolamine compounds do not fragment the same way as the phosphocholine compounds.

### 1.4. Steroid Ethanolamine Phosphodiesters (SEP)

SEP were detected in adrenal and ovarian extracts, but were not in serum [7]. On the basis of this localization, SEP seem to be precursors, rather than metabolites, of phosphocholine steroids. Apparently, the SEP are storage forms and are converted to the phosphocholine diesters by N-methylation. N-methylase is an ACTH-dependent enzyme, and also converts norepinephrine to epinephrine. Thus, in response to ACTH, [a] norepinephrine could be converted to epinephrine, leading to increased glycolysis, and [b] SEP could be converted to a phosphocholine steroid, leading to increased potassium in the heart, increased calcium, and increased blood flow. This combination of events would be important in both stress response, and during parturition [9].

### 1.5. Mass Spectral Analysis of Steroid Ethanolamine Phosphodiesters (SEP)

SEP have a different fragment pattern on mass spectrum analysis, compared to that observed with phosphocholine steroids. As there is no trimethylamine, SEP cannot fragment with loss of trimethylamine fragment—the Type A cations. Instead, the S fragment ions for SEP are formed in higher relative intensity. Whereas the phosphocholine fragment at *m*/*z* = 184 Da is a major ion in spectra from phosphocholine steroids, SEP do not generate the corresponding fragment at *m*/*z* = 142 Da.

## 2. Biochemistry of Spiral Steroids

In the 1950s, cardiotonic glycosides were proposed as a substitute for an endogenous potassium sparing hormone [1], and a synthetic potassium sparing diuretic, spironolactone, was discovered [11]. However, no one was successful in preparing a tissue extract that had the same function as spironolactone. In the 1970s, to assist physicians in monitoring the possible toxicity of digoxin, an RIA for digoxin was developed [12]. From time to time, prior to initiation of therapy, patients were identified who had unknown materials detected by the assay [13]. The material was given a trivial name, digoxin-like-material, which was abbreviated as DLM.

HPLC chromatograms are monitored by the mass ions characteristic of C341. The takeaway finding from Figure 2 and Figure 3 is that all of the expected ions characteristic of C341 co-eluted, and there were no significant unexplained ions. In total, the two Figures suggest a high degree of purity for C341.

Figure 2 and Figure 3 from [7].

This is the mass spectrum of C341, as shown on line 5 of Table 1. Where: [i] S is the ion at 341 Da; [ii] A is the ion at 487 Da, generated by loss of trimethylamine from C; [iii] B is the ion at 524 Da as the H+ ion; [iv] C is the ion at 546 as the Na+ ion Da; and [v] D is the ion at 562 Da as the K+ ion. The Ion at 503 Da is the loss of trimethylamine from D.

### 2.1. Initial Candidates for DLM

Although the levels of DLM are undetectable (less than 0.05 ng/mL) in sera from normal individuals, there are more than 200 papers describing DLM levels in various diseases [14]. Several teams attempted to isolate DLM. First, there were claims for digoxin [2]. Secondly, Hamlyn isolated 13 µg of ‘ouabain’ from 80 L of plasma [3]; there is now a long list of papers describing Endogenous Ouabain (EO) [15,16], and a shorter list describing EO as fantasy [17]. A third candidate was marinobufagenin [18]; there is an RIA for marinobufagenin that detects an unidentified substance in human serum from pregnant women [19]. It should be noted that these three cardiotonic steroids are poisons and, in human serum samples, have only been detected by RIA; no precursors are known, and there are no logical biosynthetic pathways proposed in mammals.

### 2.2. How Did We Find Spiral Steroids?

Our discovery path began with two investigations in the 1980s. First, Chasalow laboratory observed that patients with Smith-Lemli-Opitz Syndrome (SLO) had [a] a steroid disorder, [b] were potassium wasting, and [c] benefited from digoxin therapy [20]. Secondly, Bradlow laboratory observed that some human breast cyst fluids had high potassium levels [21]. Together, we speculated that the cyst fluids had a steroid hormone that infants needed, but did not make. Chasalow measured ‘digoxin’ in both types of samples [22]. Whatever the material detected with the RIA for digoxin was, a DLM was present both in breast cyst fluids and in serum from healthy newborn infants, but it was not present in the serum from newborn infants with SLO syndrome [20].

### 2.3. Why Were Spiral Steroids Not Found Prior to These Investigations?

The DLM levels in many breast cyst fluids were over 0.6 ng/mL [22]. As part of normal patient care, the fluids were aspirated and this was the initial source of the efforts to isolate DLM. First, we confirmed that the DLM was not digoxin (or ouabain) by difference in solubility. In the second stage, we learned to isolate and concentrate DLM. This could not be done earlier as methods to isolate either digoxin or classical steroids cannot be used to concentrate DLM. As the properties of DLM are different from that of all other steroids, new methods were also needed for purification [7].

### 2.4. How Did We Isolate the Phosphocholine Steroid Diesters?

The first source we used to isolate the DLM was breast cyst fluid with high potassium levels. Separation was achieved by HPLC, with a gradient of increasing polarity. Repeated chromatography led to two compounds: (a) a compound with a UV absorption at 240 nm, that was not a DLM; and (b) a DLM with little UV absorbance over 210 nm. However, as the average volume of fluid in a human breast cyst is less than 2 mL, this was difficult to use as a source for isolation of an unknown steroid-like material.

In the second stage, we obtained outdated blood plasma from the local blood collection center and, with the methods developed for the cyst fluid extract, we were able to distinguish four steroid phosphodiesters. In the third stage, we obtained 10 L of porcine blood. In total, with that starting material, we obtained about 10 mg each of four pure phosphocholine steroids diesters—C313, C337, C339, and C341. MS ion peaks for the four compounds are listed in Table 1 [7].

### 2.5. Determining the Chemical Formula of the Steroid

Chasalow applied a trial-and-error (T&E) method to determine the molecular formula of the steroid; there is no a priori reason why only one formula would satisfy the basic rules of chemistry. Once there was a logical composition, and with the knowledge of steroid biochemistry, it was possible to propose a structure. Appendix A shows the T&E analysis of each phosphodiester, its proposed composition, and the structure most consistent with steroid enzyme biochemistry. Table 2 summarizes the various features. The proposed structures do not eliminate isomers and stereoisomers. If Occam’s Razor does not apply, then there would have to be more unknown intermediates, enzymes, and pathways. For example, C313 and C329 differ by 16 Da, as would be expected for an extra hydroxy group. This was confirmed by fragmentation showing loss of 18 Da for C329. It does not identify the site of the hydroxy group, but carbon atoms 18, 19, and 21, are eliminated, as a hydroxy group on those carbons would not fragment by loss of a molecule of water (18 Da).

### 2.6. Biosynthesis of Steroid Ethanolamine Phosphate Diester—Part 1—The Phosphate

Both choline and ethanolamine are required nutrients in mammals [23]. The first step is the condensation of 17α-hydroxy-pregna-5,7-dienolone with a phosphate donor (See Figure 4).

There are, basically, two alternatives: [a] a transfer from either a phosphocholine lipid, or a phosphoethanolamine lipid; and [b] de novo, in situ synthesis, perhaps from CDP-serine or phosphatidyl-serine. The serine-steroid phosphodiester would be an anion, and would not be detected in the cation mass spectrum. Decarboxylation would form E313 [9] and N-methylation would convert E313 to form C313 [24]. As we detected many Exxx compounds in adrenal extracts, methylation might not occur until after the addition of the extra carbons. This would fit with the hypothesis of steroid phosphoethanolamine diesters as the precursor-storage form for the phosphocholine diesters (Figure 3) [7].

### 2.7. Biosynthesis of Phosphosteroid Diesters—Part 2—The Steroid

In 1940, Thomas Wolfe wrote: “You can never go home again.” That same principle applies in steroid biochemistry. You cannot reform a Δ7-8 sterol once it has been reduced by the Δ7-sterol reductase. UV spectra of both of the steroid phosphocholine diesters with 21 carbon atoms (C313 and C329) are consistent with Δ5-Δ7-dienes. Thus, patients with SLO syndrome (7-dehydrosterol reductase deficiency) provide the ‘tell’ that the precursor for the phosphodiesters must be a Δ7-sterol.

After almost a century of steroid investigation, no unconjugated steroids with more than 21 carbon atoms have ever been described in vertebrates. One could visualize two basic paths to synthesize steroids with more than 21-carbon atoms, forming them with: [a] a different side chain cleavage enzyme from cholesterol; and [b] adding carbons to a precursor with 21-carbon atoms. With side chain labeled cholesterol, Burstein found only a 6-carbon fragment [25], which confirmed that these steroids had to be derived from a 21-carbon precursor. When only phosphocholine steroids with 23-carbon atoms had been identified, the extra carbons could have been added by malonyl coenzyme A (and subsequent decarboxylation).

Fatty acid synthase lengthens carboxylic acids, 2-carbons at a time. Acetyl coenzyme A is the donor. After condensation, it is dehydrated and the resulting alkene is reduced. Figure 1 shows how 23-carbon atom phosphosteroids could be synthesized by a fatty acid synthetase-like enzyme. However, that does not account for the 24 and 25-carbon atom phosphosteroids.

The three most common acyl coenzyme A derivatives, and the corresponding phosphocholine steroids, are listed in Table 3. It is possible that other acyl coenzyme A derivatives could also be coupled to C313 or C329. This would depend on the substrate specificity of the fatty acid synthase-like enzyme complex.

Black boxes: precursor steroids with 21-carbon atoms; Red boxes: proposed intermediates; Blue boxes: spiral steroids; Green boxes: carboxylic acids. C357* is the branch point. The intermediate steps are similar to the steps leading to lengthening of fatty acids. Ring closure leads to Ionotropin (C341) and DLM. Reduction of the C20-C23 alkene prevents ring closure. The function of C361* and C363* are unknown at this time.

The extra carbons are added by the action of an unknown enzyme catalyzing a process similar to that of fatty acid synthase, leading to C357*. C357* is a substrate for two different enzymes: [i] Ring E closure enzyme, leading to spiral steroids; and [ii] C20-C23 reductase, leading to carboxylic acid compounds. The dialkenes of both compounds can be sequentially reduced, first at the Δ7-8 alkene, and then at the Δ5-6 alkene. The final product is a 5β-steroid. The Δ5-6 intermediate may also have a function.

Fatty acid synthase enzyme complex also seems to accept propyl Coenzyme A (which leads to a steroid with 24-carbon atoms), and acetoacetyl-Coenzyme A (which leads to a steroid with 25-carbon atoms), as a co-substrate with X313.

Black boxes: precursor steroid with 21-carbon atoms; Red boxes: proposed intermediates; Blue boxes: spiral steroids. The intermediate steps are similar to the steps leading to the lengthening of fatty acids. C369 fragments with the loss of 18 Da, which is consistent with the presence of a hydroxyl group. For C369 and C413, we have no evidence indicating at which step the hydroxyl groups were added, nor the carbon atoms to which they were attached. They are shown at carbon-11 and carbon-16 as enzymes with that specificity are known.

## 3. Proposed Function of Steroid Phosphocholine Diesters

Regulation of potassium recovery must occur in every tissue; however, there must be a source of specificity that causes accumulation in only one target tissue. The primary function of the steroid type of potassium sparing diuretics is regulation of the NaK-ATPase, of which there are multiple isoforms. We propose that these match with different classes of steroid phosphodiesters.

Four of the steroid phosphodiesters are likely to have unique functions. Each class has a spiral steroid. Some of the classes also have members with a hydroxy group. The 23-carbon atom class also has a carboxyl derivative that does not seem to be a precursor, and was present in the adrenal extracts. The following sections correlate each class with their proposed endocrine function.

### 3.1. Function of the 23-Carbon Atom Class of Steroid Phosphocholine Diesters

Ionotropin (C341) is the major 23-carbon atom steroid phosphocholine diester. It was isolated on the basis of cross-reaction with digoxin-specific antibodies. The pharmaceutical equivalent, spironolactone, improves renal and heart function in patients with congestive heart failure. ‘Endogenous ouabain’, which was also isolated on the basis of cross-reaction with ouabain-specific antibodies, was elevated in patients with essential hypertension [26,27]. The actual compound measured is probably C341; however, whether C341 is the cause of the essential hypertension, or part of the attempt to compensate for the hypertension, remains to be determined.

### 3.2. Function of the 24-Carbon Atom Class of Steroid Phosphocholine Diesters

The 24-carbon compounds are C353, C369, and E389*. Steroids of this class were not detected in serum from prepubertal children. E389* was present in bovine adrenal extracts; C369 was present in bovine ovaries; and C353 was present in serum from pregnant women. Based on where they were found, this class probably binds to the specific NaK-ATPase isoforms that are present in gonads, and in the fetal-placental unit. Elevated levels of C313 and/or of C329 in serum seem to be a common feature of women with pre-eclampsia [28]. Thus, when fetal potassium is inadequate the placenta secretes both precursors into maternal circulation, leading to potassium accumulation and maternal hypertension [29]. C369 was present in the serum of obligate heterozygotes for 7-dehydrosterol reductase deficiency [30]. This suggests that C369 (11-hydroxy, 23-methyl-C339) does not cause hypertension; thus, the theory is that hypertension is caused by the C341 made from high levels of C313 present in maternal serum. Elevated C329 could lead to synthesis of C369, and lead to restoration of fetal potassium levels [31].

### 3.3. Function of the 25-Carbon Atom Class of Steroid Phosphocholine Diesters

Figure 2 also describes the proposed biosynthetic path for the synthesis of steroids with 25-carbon atoms. C381 is the most common member of the 25-carbon atom class. It was found in milk from cattle, goats, and sheep. Milk from all three species have high levels of potassium, and it was also present in pregnant women, whether or not they had pre-eclampsia [28]. C413 was detected in fetal calf serum. In addition to binding to NaK-ATPase, it is possible there is a specific nuclear receptor in mammary glands for a steroid of this class. C381 was also detected in serum from adult males; its function in males is less obvious.

### 3.4. Spiral Steroids during Pregnancy

Figure 5 shows the representative mass spectra obtained on serum extracts from pregnant women [28]. The study included 40 samples, 20 samples from women with normal blood pressure, and 20 samples from women with pre-eclampsia. Miltefosine (sodium hexadecyl phosphocholine) generates an ion at *m*/*z* = 430 Da. As an internal control, 10 µL of 0.2 mg/mL of miltefosine was added to 0.2 mL of serum. For analysis, the ion intensity of each steroid phosphodiester was compared to the intensity of the internal control. If the peak ion intensity of the phosphoester was the same as that of miltefosine, then the concentration of the phosphodiester would be about 10 mg/L (~20 µM) of serum. In fact, when C341 was isolated, 10 mg was isolated from 10 L of porcine blood [7]. Note that in the spectrum from normotensive woman, ion intensity of C341 was about 1/5 the intensity of the miltefosine ion.

During pregnancy, all four classes of steroid phosphodiesters are present in serum: (a) 21-carbon class: C313 and C329; (b) 23-carbon class: C341; (c) 24-carbon class: C353; and (d) 25-carbon class: X381. There are three different spiral steroids present in maternal serum; hence, it is not surprising that a single measurement with an immunoassay specific for a cardiotonic glycoside, such as ouabain or digoxin, does not adequately reflect fetal-maternal health status.

Peak assignments: (i) 353 Da—C353; (ii) 381 Da—X-381; (iii) 430 Da—Internal control, miltefosine; (iv) 475 Da—Type A ion from C329; (v) 518 Da—type C ion from C313; and (vi) 546 Da—Type C ion from C341.

### 3.5. Is There a Role for Replacement Hormone Therapy for Spiral Steroids?

Patients who benefit from spironolactone and/or eplerenone would be expected to benefit from one of the spiral steroids. The synthetic compounds probably bind to all of the isoforms of NaK-ATPase. Potentially, natural spiral steroids may be tissue specific and have fewer side effects.

## 4. Materials and Methods

This review tries to summarize the discovery of these novel steroids. The details for the methods of extraction are in Chasalow [7]. Basic methods for the mass spectroscopy of individual serum samples are in Chasalow, Bochner, and John [28].

## 5. Conclusions

Aldosterone was discovered in 1955, and even 70 years after their discovery, investigators continue to learn more about the role of mineralocorticoides. However, the clock has now started ticking on the biochemical endocrinology of the spiral steroids. The problem is complicated by the fact that, whereas aldosterone functions in a similar fashion in all of its targets, the same is not true for spiral steroids. C341 (Ionotropin) functions in renal and cardiac tissues; C369 works in the gonads and placenta; and C381 accumulates potassium in the milk of mammary glands. These three compounds are spiral steroid phosphodiesters, and they bind to different forms of NaK-ATPase. It is as if we considered estrogens, androgens, and progestogens as all functioning in the same way just because they all bind to nuclear receptors, even though the receptors are all different.

## Data Availability

There is no new data presented in this article.

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
