# Peer review of "An Introduction to Spiral Steroids"

_ijms, 2022, doi:10.3390/ijms23179523_

Round 1

Reviewer 1 Report

This is a good introductory review on spiral steroids, including its chemical structure and features, discovery, location and function.

a figure to show the structure difference between normal steroid and on a typical spiral steroid

Line 30-31 states “. Some, but not all of them, bind to digoxin specific antibodies”, provide more detail on this in a table or in legend of Table 2.

Use a table or figure to list all spiral steroids that have been discovered, including isolation method, tissues, level and/or related diseases

Author Response

Reviewer #1

An extra figure – Scheme 1 and Scheme 2 both have figures showing individual steroids.

Line 30-31 – This review is not about digoxin specific antibodies. Many scientists have generated antibodies to digoxin but each one is different. In order to determine if an unknown compound will bind to a specific antibody that also binds to digoxin, we would have to purify each of the unknown compounds. Six compounds -marked with ‘&’ – were so purified. The digoxin specific antibody we used detected the three spiral steroids but not the other three compounds. We have not investigated any other antibodies with specificity for digoxin.   

Table 1 lists ALL of the spiral steroid phosphodiesters that that we discovered. I know of no other steroid phosphodiester or spiral steroids discovered by other investigators.

Page 4 with Table 1 may have been rearranged in the copy you received for review.

Reviewer 2 Report

This is a comprehensive review on the steroids with 7-dehydraocholesterol, along with their identification and function. The author has led several important studies in this field and is well-positioned for this review article. A minor comment is about the figures and tables. There are 2 figure 4, one on line 194 and one on line 314. In addition, it seems like figure 3 and table 1 is describing the same chart. Please have it clarified and keep one title, figure or table.

Author Response

Response:

Line 314 and 297 should have been designated as Fig 5 -  correction made.

The manuscript was inadvertently disorganized. Table 1 was inserted between Figure 3 and its legend. This has been corrected. 

Reviewer 3 Report

The paper is well written to be a primer reviewer for early career scientist to learn about spiral steroids. I have recommendation just to improve the figures and tables throughout the manuscript. 

For figure 1 please change the background to be white it makes it easier for the readers to assess the structures especially if the reader chooses to print the copy. 

I recommend that figure 2 and 3 be an actual digital figure instead of a scanned image from a paper, since it seems like the citation provided in newer. 

Table 1 and 2 be reformatted to match Table 3 in terms of fonts etc. 

Figure 4 : the structures can be cleaned up to show a reasonable bond angle across all bonds, a quick chem drawing software clean up should do this. 

Similarly for Scheme 1 

Otherwise small spell errors that can be corrected and a few extra spaces left here and there. 

The paper otherwise is well written and should be beneficial to the community. 

Author Response

Figure 1 – My copy of the software doesn’t make white background.

The original figures from 2 and 3 are no longer available.  (Two moves is as good as a fire.)

I have increased the size of the Tables to the max permitted by the journal format.

Chem drawing did not correct the bond angles AND it  rotated the figure.

I ran spell check again and corrected a few typos.